# UKAN: UNBOUNDED KOLMOGOROV-ARNOLD NETWORKS

## ABSTRACT

We present Unbounded Kolmogorov-Arnold Networks (UKANs), a novel algorithm that eliminates the need for bounded grids in traditional Kolmogorov-Arnold Networks (KANs). The key innovation is a coefficient generator (CG) model that dynamically produces B-spline coefficients, operating on an infinite symmetric grid. UKANs integrate multilayer-perceptrons with KANs, using positional encoding of grid groups as input to the CG model. This approach enables function approximation on unbounded domains without data normalization. Additionally, to reduce UKAN and KAN computational cost, we introduce a GPU-accelerated library that reduces B-spline evaluation complexity by a factor of $\mathcal{O}(\text{grid size})$ compared to existing libraries, enabling efficient large-scale learning. Our experiments on regression, classification, and generative tasks demonstrate UKANs' effectiveness, while benchmarks confirm superior memory and computational efficiency compared to existing methods. This work advances function approximation techniques, offering a flexible solution for complex, large-scale learning problems.

## 1 BACKGROUND

Neural networks (MLPs) are the workhorse of the current AI and Deep Learning revolution, driving advances in computer vision, language models, computational science, and more recently biology and molecular science LeCun et al. (2015); Goh et al. (2017); Schütt et al. (2018); Pandey et al. (2022); Raissi et al. (2017). The universal approximation theorem guarantees that MLPs with enough parameters can fit any function. The widespread adoption of MLPs across various disciplines has led to the emergence of exciting applications such as ChatGPT in LLM and AlphaFold in protein structure prediction Vaswani et al. (2017); Jumper et al. (2021). However, MLPs suffer from a few drawbacks, particularly generalization for regression tasks.

Recently, the Kolmogorov-Arnold network (KAN) Liu et al. (2024) has gained attention as a promising alternative to traditional MLPs, especially in scientific applications, with various variants currently under development Bozorgasl & Chen (2024); Genet & Inzirillo (2024); Abueidda et al. (2024); Kiamari et al. (2024). The KAN architecture is partially based on the Kolmogorov-Arnold representation theorem Kolmogorov (1961); Braun & Griebel (2009), which states that any multivariate function on a bounded domain can be obtained by a finite composition of continuous univariate functions and summation. Mathematically, this can be represented as:

$$f(x) = \sum_{q=1}^{2n+1} \phi_q \left( \sum_{p=1}^{n} \phi_{q,p}(x_p) \right) \tag{1}$$

where $\phi_{p,q} : [0,1] \rightarrow \mathbb{R}$ and $\phi_q : \mathbb{R} \rightarrow \mathbb{R}$. For a more detailed explanation of the KAN architecture and mathematics, we refer eager readers to the original KAN paper Liu et al. (2024).

Despite their potential, practical use of KANs is currently hindered by compute and memory inefficiencies, which stems from implementation challenges and the partial reliance on grid update tricks. Without grid update tricks, KAN learning stagnates as inputs into the layer can fall outside of the grid domain fixed at the beginning of training. Additionally, the numerical instability of the grid update implementation prevents the original KAN from fully leveraging the capabilities of graphical

Table 1: Compute complexity of Torch- and Warp-KAN for single layer B-spline evaluation.

| Model | Complexity |
|-------|-----------|
| Torch KAN | $\mathcal{O}(K d_g d_{in} d_{out})$ |
| Warp KAN | $\mathcal{O}(K d_{in} d_{out})$ |
| Warp UKAN | $\mathcal{O}(K d_{in} d_{out}) + \mathcal{O}_{CG}(d_{emb}^2 + d_{emb} d_{out} K)$ |

processing units (GPUs), which are driving current AI advancements. Another drawback of KAN is its inability to leverage batch computation due to the B-spline evaluation being implemented with a complexity of $\mathcal{O}(k d_g d_{in} d_{out})$, where $k$, $d_g$, $d_{in}$, and $d_{out}$ are the degree of B-spline and grid, input, and output dimensionalities, respectively. However, this computational complexity issue is resolved in this work for the original KAN. We note that B-splines have been used in other works before KAN to learn activation functions, and our library can also be used to accelerate them Bohra et al. (2020). Furthermore, our library facilitates a rigorous and unbiased comparison between KANs and MLPs in terms of floating-point operations (FLOPs), thereby bridging the gap between the typically naive implementations of KANs and the extensively optimized implementations of MLPs that have been refined over decades Yu et al. (2024).

## 2 ALGORITHM

The solution to compute and memory issues of KANs lies in exploring the local nature of B-splines instead of the recursive formula over the entire grid. In other words, the evaluation of the B-spline function is local and depends on the $k + 1$ coefficients, leveraging this observation, we represent the B-spline function with basis matrices Qin (1998) as shown in Equation 2.

$$P(u) = U\mathbf{M}P \tag{2}$$

where $u = x/\delta g - \lfloor x/\delta g \rfloor$ and $\delta g$ is the distance between two adjacent grid points. $U$ is the vector of $(1, u, u^2, ..., u^k)$. $\mathbf{M}$ is basis matrix and $P$ is the vector of B-spline coefficients $(p_0, p_1, ..., p_k)$. The basis matrices are obtained by applying recursive B-spline equations, and only depends on the degree of the B-spline function, as commonly used in other disciplines Moradzadeh et al. (2018); Mashayak et al. (2015). For a B-spline of order 3, the B-spline evaluation can be written as,

$$P(u) = \begin{bmatrix} 1 & u & u^2 & u^3 \end{bmatrix} \frac{1}{6} \begin{bmatrix} 1 & 4 & 1 & 0 \\ -3 & 0 & 3 & 0 \\ 3 & -6 & 3 & 0 \\ -1 & 3 & -3 & 1 \end{bmatrix} \begin{bmatrix} p_0 \\ p_1 \\ p_2 \\ p_3 \end{bmatrix} \tag{3}$$

We provide efficient implementations of the above formula using NVIDIA Warp Macklin (2022) in a new library called warpKAN with evaluation complexity of $\mathcal{O}(k d_{in} d_{out})$ along with PyTorch bindings Paszke et al. (2019). This implementation offers both memory and computation efficiency, as described in Table 1. However, the reduction in compute and memory cost of B-spline components does not solve the issue of bounded range of the grid in the original KAN.

To achieve an unbounded domain for the KAN grid, we use an MLP to generate B-spline coefficients, called a coefficient-generator (CG) MLP. As shown in Equation 2, every B-spline of order $k$, needs $k + 1$ coefficients ($K = k + 1$). One approach is to predict coefficient by calling $K$ times same MLP for $K$ adjacent grid indexes, however, in our experimentation we notice this algorithm fails.

To address this issue, we group $K$ adjacent grid indexes together, called a group index, and use the CG model to predict $K$ coefficients for each group index. The selection process works as follows: we first concatenate the $K$ coefficients predicted by the previous group index and the one grid index belongs to, resulting in a sequence of $2K$ coefficients. We then select the $K$ coefficients starting from index $i = g_{id} \mod K$ to $i + K$ ($g_{id}$ is the grid index). This ensures that we obtain the correct $K$ coefficients for each grid index, while minimizing the number of MLP and embedding calls. In other words, instead of embedding every grid index we embed every grid group index i.e. the collection of $K$ adjacent grid indices. The input to the CG MLP is obtained by concatenating the embedding of

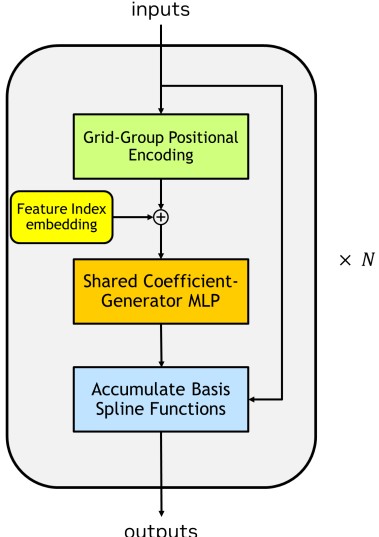

Figure 1: The UKAN - model architecture including grid group positional encoding, coefficient-generator MLP, and B-spline function.

the feature index and the sinusoidal positional encoding of the group index as used in transformers Vaswani et al. (2017). The architecture of UKAN is depicted in Figure 1. Note that there exists a one-to-one mapping between UKAN and KAN within specific grid bounds. After training, one can evaluate the CG model for all inputs and the results can be stored as KAN parameters. This transformation enables the application of all symbolic regression and pruning techniques used in KAN, preserving interpretability while gaining the benefits of UKAN during training.

## 3  EXPERIMENT

We perform several experiments to benchmark the performance of current implementations and capabilities of UKAN in regression, classification, and other tasks.

### 3.1  BENCHMARKING

In Figures 2 and 3, we benchmark warp- and torch-KAN in different setups, *i.e.:* varying grid size and order of the B-spline to evaluate the compute and memory improvements achieved in the current work and the accompanied library. Particularly, in Figure 2, we compare compute cost of the forward & backward passes, and their sum for a single layer KAN [32, 32] with different B-spline orders. During this comparison grid size and batch size of 64 and $2^{16}$ are used and all results for warpKAN are normalized with respect to the PyTorch implementation of the original paper. The results indicate a performance improvement of 5.5-15x which increases as order of B-spline increases.

In Figure 3, we benchmark the KAN [32, 32] with B-spline order of 3 on different grid sizes. A batch size of $2^{17}$ is used for this experiment, we observed that torchKAN (naive implementation of B-spline) runs out of memory for a grid size equal and larger than 256, while warpKAN can reach grid sizes of up to $2^{18}$, more than 1000x larger. By reducing the computational and memory costs of B-spline evaluation, KANs can become a more viable option for large-scale applications, paving the way for their widespread adoption beyond toy datasets.

### 3.2  TASKS

To evaluate the performance of UKAN in regression, we conducted three experiments.

    I.  $f(x, y) = \exp(J_0(20x) + y^2)$, where $J_0$ is the Bessel function of order 0.

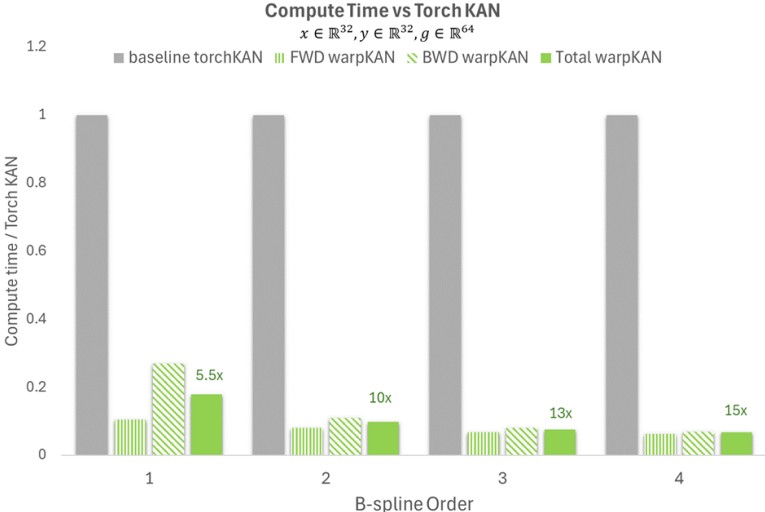

Figure 2: Performance benchmarking of current and naive implementation of KAN with different B-spline orders. We compared compute time of warp- versus torch-KAN for a single layer with different B-spline orders, (32-dimensional input and output and grid size of 64 on NVIDIA A6000 GPU and batch size of $2^{16}$). We find that warpKAN is on average 10x faster and up to 15x faster for higher order B-splines.

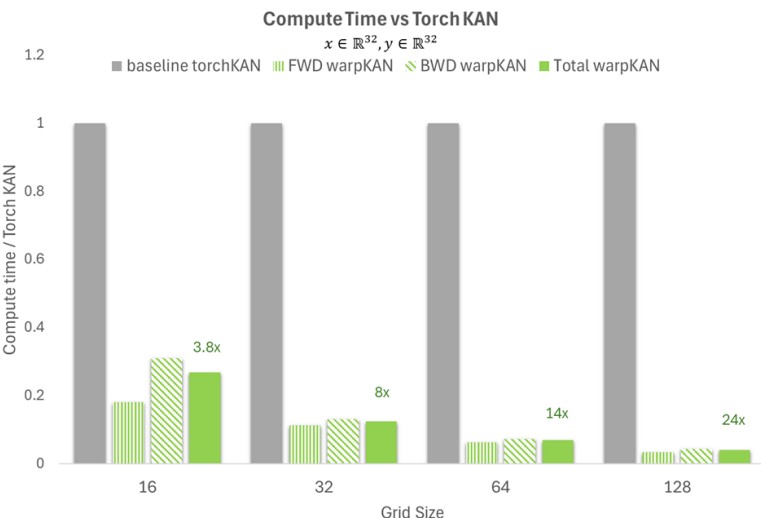

Figure 3: Performance benchmarking of current and naive implementation of KAN with different grid sizes. We compared compute time of warp KAN versus torch KAN with order 3 B-spline and 32-dimensional input and output over different number of grids on NVIDIA A6000 GPU and batch size of $2^{17}$. Torch KAN rans out of memory due to implementation limitations beyond 128, while warp KAN can reach up to $2^{18}$ grids, translating to 1000x larger grid size. We find that warpKAN is on average 12x faster and up to 24x faster for larger grid sizes.

    II. $f(x, y) = \exp(\sin \pi x + y^2)$

    III. $f(\mathbf{x}) = \exp\left(\frac{1}{15} \sum_i^n \sin\left(\left(\frac{4i}{15} + 1\right)\pi x_i\right)\right)$, where $i = 0, 1, \ldots, 15$ and is a high dimensional function compared with functions I and II.

We compare the results of UKAN, KAN and MLP [2, 5, 1] for 2D functions (functions I and II). For function III, we use UKAN, KAN, and MLP [16, 32, 1]. We used a two-layer MLP with 8- and 16-dimensional positional encodings and feature embeddings for 2D and 16D functions, respectively.

The first layer of the CG MLP uses SiLU nonlinearity and generated coefficients are scaled by another learnable parameter to improve learning, analogous to the original KAN paper. An Adam optimizer Kingma & Ba (2014) with a learning rate of 0.01 and weight decay of $1e^{-5}$ for 200,000 epochs is used to minimize the MSE loss. The learning rate is decayed exponentially with the rate of $1 - 1e^{-4}$ and minimum learning rate of $1e^{-4}$. The results are shown in Figure 4, where UKAN and KAN perform much better than MLP, and KAN performs better than UKAN. In theory, KAN and UKAN have the same learning capacity, but the MLP component of UKAN might slightly hurt generalization and performance compared to KAN. One might switch to a KAN-based CG model instead of an MLP-based CG model to resolve the generalization issues, but this comes with large computational costs. In terms of a fixed FLOPs budget, our study provides a more robust and fair comparison in terms of KAN and MLP accuracy, even though this was not a target of our study. We additionally note that although the compute cost of a KAN is greater than that of an MLP by a factor of $K$, KAN convergence often compensates for this factor.

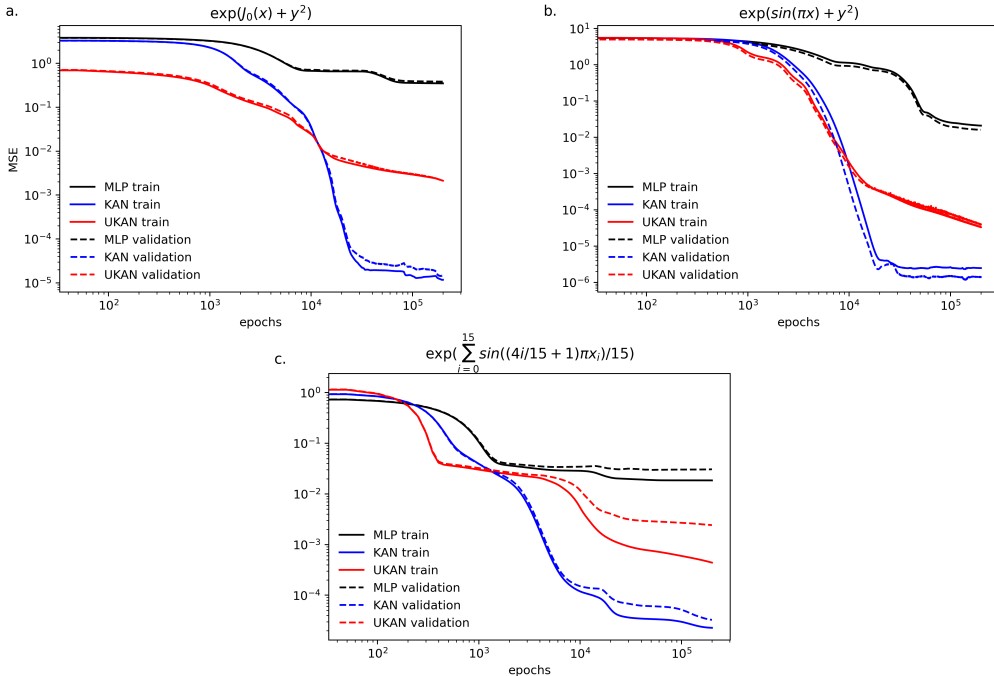

Figure 4: The regression task results. a. RMSE over training epochs of function I with KAN, UKAN, and MLP [2, 5, 1]. b. RMSE over training epochs of function II with KAN, UKAN, and MLP [2, 5, 1]. c. RMSE over training epochs of function III with KAN, UKAN, and MLP [16, 32, 1].

KANs promise better generalization compared to MLPs for regression tasks, similar to equivariant models allowing for the exploitation of symmetries for improved generalization. In particular, E(n)-Equivariant Graph Neural Networks (EGNNs) are equivariant with respect to the translations, rotations, and permutations Satorras et al. (2022). Here, we explore how combining equivariance with KAN leads to improved performance in the study of n-body systems as described in the EGNN paper Satorras et al. (2022). To evaluate this, we replace the final scalar predicting MLPs in EGNN with UKAN and KAN layers. Specifically, the scalar outputs of $\phi_x$ and $\phi_v$ in Equation 4 are predicted with UKAN and KAN.

$$\mathbf{v}_i^{l+1} = \phi_v(\mathbf{h}_i^l)\mathbf{v}_i^{init} + C \sum_{j \neq i} (\mathbf{x}_i^l - \mathbf{x}_j^l)\phi_x(\mathbf{m}_{ij}^{l+1})$$

$$\mathbf{x}_i^{l+1} = \mathbf{x}_i^l + \mathbf{v}_i^{l+1}$$

(4)

We keep the rest of parameters and datasets identical to the original paper and their code on Github. We also train the SE(3) Transformer model as another reference point. The results are shown in Table 2, where we observe that UKAN and KAN improve the accuracy compared to the original architecture, and the improvement of UKAN is better than the KAN model.

Table 2: Mean Squared Error for the future position prediction in the N-body system.

| Method | MSE |
|---|---|
| EGNN | 0.00638 |
| EGNN+KAN | 0.00609 |
| EGNN+UKAN | **0.00591** |
| SE(3) Transformer | 0.02469 |

We explore the effectiveness of UKAN and KAN in physics-informed neural networks Karniadakis et al. (2021) to solve the logistic growth model, which is used to model population dynamics in biological and ecological systems. For this experiment, we use both UKAN and KAN [1, 5, 1] without MLP component to solve the differential equation below,

$$\frac{df}{dt} = Rf(t)(1 - f(t)) \tag{5}$$

where $R$ is the growth rate set to 1.0 and the function $f(t)$ represents the growth rate of the population over time (t). We impose boundary condition of $f(0) = 0.5$ to uniquely specify the solution and compare the results with the analytical solution of $f(t) = \frac{1}{1+\exp{(-t)}}$. We use domain of $[-5, 5]$ to sample data and Adam optimizer with the learning of rate of $1e^{-3}$ and weight decay of $1e^{-5}$ and follow the standard procedure for PINN minimization, i.e. minimizing MSE of the differential equation residual over colocation points and boundary conditions. The results are shown in Figure 5, for and KAN [1, 5, 1] without the MLP component. UKAN and KAN acheive MSE of $1e^{-5}$ and $1e^{-6}$, respectively on the sample dataset, indicating both models are applicable to physics-informed neural networks scenarios.

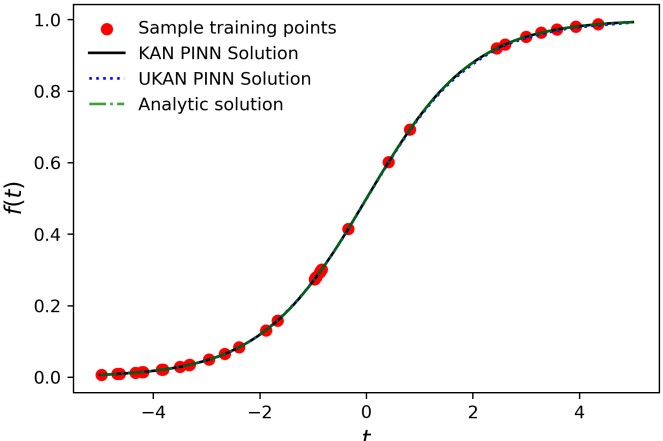

Figure 5: KAN and UKAN used in PINNs. Solving logistic growth model with both KAN and UKAN [1, 5, 1] over domain of -5 and 5. Both models match the analytical solution with good approximation.

We devised two experiments to evaluate UKAN's performance in classification tasks. In particular, we trained UKAN and KAN over Moon- and MNIST-datasets. We report the accuracy of training and validation of both models on the moon-dataset in Table 3, where we observe close to 100% accuracy for Moon-dataset with 2D inputs with slight superior accuracy of UKAN over KAN in this setup. Both UKAN and KAN [2, 4, 2] are trained using SGD optimizer with learning rate of 0.01 for 10000 epochs, and results are averaged over 3 different initializations.

Table 3: Moon dataset classification accuracy.

| Model | Training | Validation |
|-------|----------|------------|
| KAN | $98.46 \pm 1.3$ | $98.53 \pm 0.4$ |
| UKAN | $\mathbf{100.0 \pm 0.}$ | $\mathbf{99.83 \pm 0.17}$ |

The final classification task was performed on the MNIST dataset, where we trained both the UKAN and KAN models with configurations [784, 32, 10] and a degree-3 B-spline. Both models were optimized using the Adam optimizer combined with an Exponential scheduler, having a learning rate of $2 \times 10^{-4}$ and a decay rate of $1 - 10^{-4}$. The KAN network incorporated 51 grid points across the interval $[-10, 10]$, whereas UKAN utilized a grid delta of 3.0 and a 48-dimensional positional encoding. Notably, both models employed only the B-spline component without any MLP components.

Figure 6 illustrates the learning curves for both the training and validation datasets. Training was halted upon detection of overfitting in the training dataset. Furthermore, three rounds of independent training with different initializations were conducted to compare the performance of UKAN and KAN. The results, as presented in Table 4, indicate that UKAN outperforms KAN on the validation dataset while slightly underperforming on the training dataset.

Table 4: MNIST dataset classification accuracy.

| Model | Training | Validation |
|-------|----------|------------|
| KAN | $\mathbf{98.93 \pm 0.78}$ | $95.35 \pm 0.04$ |
| UKAN | $98.40 \pm 0.24$ | $\mathbf{96.29 \pm 0.08}$ |

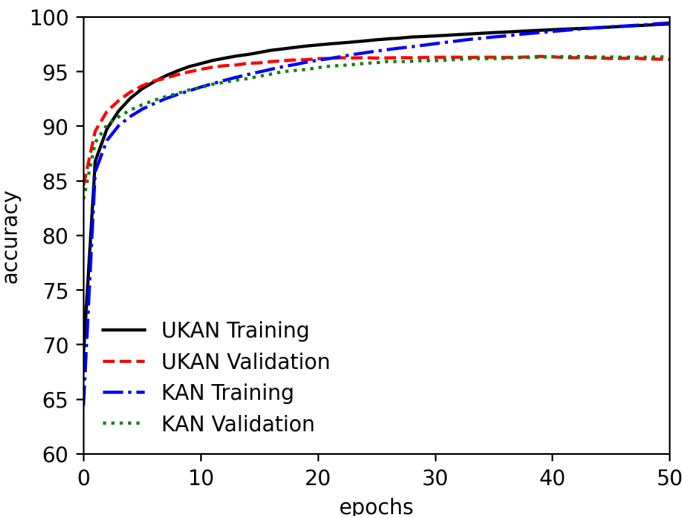

Figure 6: Classification task over MNIST dataset. Accuracy of training and validation datasets for UKAN and KAN [784, 32, 10] as training progresses. UKAN uses a 48-dimensional positional embedding and feature index embedding.

Finally, we evaluated the performance of three different architectures for Denoising Diffusion Probabilistic Models (DDPM) Ho et al. (2020) on a synthetic 2D circle dataset with added noise. The first architecture is only composed of MLPs, while other architectures use KAN and UKAN in the input of temporal layers and output layer (see Appendix A for full architecture). We used Adam optimizer with a learning rate of $5e^{-5}$ for 500 epochs with a batch size of 800. Our results demonstrated that both KAN and UKAN significantly outperformed MLP in terms of the Wasserstein distance shown in Table 5 and sample quality as shown in Figure 7. Data samples from original distribution and generated from DDPM with KAN, UKAN, and MLP architectures indicates superior performance of

KAN and UKAN compared to MLP and slightly superior performance of UKAN over KAN. This result indicates possible applications of KAN and UKAN in generative tasks, where MLP alone might fail to learn underlying data distribution especially in sample quality as we observed loss values of MLP, KAN and UKAN were very small.

Table 5: DDPM with KAN, UKAN, and MLP

| Model | Wasserstein distance |
|---|---|
| KAN | 0.693 |
| UKAN | **0.655** |
| MLP | 1.058 |

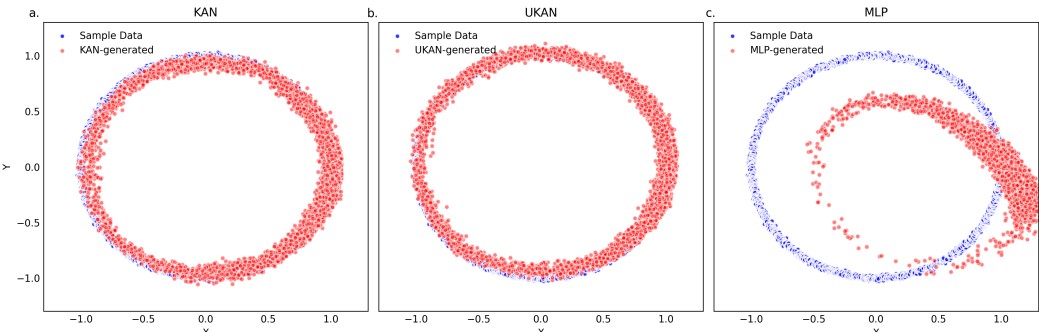

Figure 7: DDPM with KAN, UKAN, and MLP.

## 4 CONCLUSION

In this work, we presented the Unbound Kolmogorov-Arnold Network (UKAN) which unifies multi-layer perceptron networks (MLPs) with KANs along with an efficient GPU implementation of the underlying components of KANs. GPU acceleration decouples the computational cost and memory fingerprint of KAN from the grid size to the order of B-spline function by using local matrix representations of B-spline functions. In addition, our proposed UKAN architecture allows using KANs without any fixed grid range limitation by generating coefficients from a coefficient-generator MLP. We evaluated our UKAN model on regression, classification and generative tasks and compared learning efficiency and accuracy. We are excited about the potential of UKAN and its variants, as well as the applications of our accompanying GPU optimized library which alleviates memory and compute issues in existing implementations. We believe this work will enable application of accelerated KAN and UKAN in areas such as molecular property predictions, protein docking, large language models, and computer vision.

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

## A  APPENDIX

The Decoder network is designed to transform input features through a series of linear and temporal layers. Here we explain architecture without KAN or UKAN layers, *i.e.* with only linear layer and SiLU nonlinearity, and mention the differences at the end. The architecture consists of an input linear layer, three temporal layers, and an output linear layer.

### A.1  ARCHITECTURE DETAILS

The Decoder is constructed with the following layers:

- **Input Linear Layer:** The initial fully connected layer transforms the input features from the input dimension to an intermediate dimension.
- **Temporal Layers:** A series of temporal layers; specifically designed for the handling of time-dependent data. In our implementation, we use three temporal layers.
- **Output Linear Layer:** The final fully connected layer transforms the intermediate features back to the original input dimension.
- **Nonlinearity:** The intermediate features passed through SiLU non-linear activation function before being processed by the temporal layers.

The Temporal Layer is designed to integrate temporal information into the feature transformation process. This layer receives the input features and a temporal embedding, processes them through a series of linear transformations, and combines the outputs with a skip connection to ensure that the temporal information is effectively incorporated.

The Temporal Layer consists of the following components:

- **Fully Connected Layers:** These layers perform linear transformations on the input features.
- **Temporal Encoding:** This layer projects the temporal embedding to the same dimensional space as the output features.
- **Skip Connection:** If the input and output features have the same dimension, an identity mapping is used. Otherwise, a linear transformation is applied to match the dimensions.
- **Output Linear Layer:** This layer produces the final output by combining the transformed features with the skip connection.

Within KAN and UKAN architectures, we only replaced the output linear layers of the Decoder network and Temporal layers with UKAN and KAN layers. We used UKAN with grid delta of 0.4 and 24 dimensional positional encoding and KAN with 11 grid between -2 and 2. Both KAN and UKAN were order 3 B-spline functions without MLP component.

