# OpenReview forum: "UKAN: UNBOUNDED KOLMOGOROV-ARNOLD NETWORKS"
_ICLR.cc/2025/Conference — Submitted to ICLR 2025_

### Official Review · Reviewer_YkyW · 2024-11-01

**Soundness:** 3
**Presentation:** 2
**Contribution:** 2
**Rating:** 5
**Confidence:** 4

**Summary:**

The paper presents a method to efficiently compute a node in KAN (Kolmogorov-Arnold Networks) using B-splines allowing unbounded coefficients that are generated using an MLP; KANs differ from MLPs by allowing non-linear operations on edges instead of nodes. They provide a GPU implementation that speeds up earlier methods by a factor of “grid size” (number of grid points for discretization). They conduct a number of experiments that show significant speed up of their method over prior methods. Experiments on several datasets including N-body problem and MNIST show improved performance over standard KAN and MLP architectures.

**Strengths:**

Provides a method to significantly speed up inference using KANs. Since KAN nodes are more expressive, this has the potential to be widely applicable.

**Weaknesses:**

The presentation can be improved. For someone not familiar with KANs, the notions of grid size, and other parameters should be explained clearly. The speed up and accuracy tradeoffs could be highlighted early on to give a sense of the impact. State what delta and g are in eqn 2.

The method used for speeding up is somewhat straight forward.

It would have helped to include some of the popular LLM style tasks in your experiments clearly showing the speed vs accuracy including standard MLP based implementation of LLMs.

**Questions:**

How exactly does your method compare with standard MLP based architectures -- do you ensure the total compute remains the same?
How would your method compare on LLMs for language prediction tasks? Do you think it will provide a speed up over LLMs?
For the setup in figure 4, where you compare UKAN, KAN, MLP how do the latency/speed of each technique compare?

---

> ### Author Response · Authors · 2024-11-18
> **Response to Reviewer 3 (YkyW)**
>
> We thank the reviewer for their comments and valuable suggestions. Below we address reviewer's concerns:
>
> # Questions
>
> Regarding implementation and comparisons:
>
> Our methodology maintains consistent node counts across MLP, KAN, and UKAN implementations.
>
> Theoretical analysis shows that KAN and UKAN operations are (order + 1) times more compute intensive than MLP compute of the input and output dimensions. However, depending on whether the implementation is memory or compute bound, this can translate into different relative performance. For example with 16 dimensional input and output and order 3 BSpline, our optimized KAN layer is 9-30x slower than MLP with the same number of nodes. Notably, our implementation is 5-120x faster than publicly available code (pykan or fastkan). UKAN is 4-12x slower than the KAN layer, but it is addressable with further development. Current performance differences are primarily due to implementation maturity rather than fundamental limitations. We plan to:
> 1. Fuse the MLP component with the B-spline component
> 2. Implement additional optimizations to improve I/O for KAN performance, cache coefficients in shared memory
> 3. Develop specialized libraries to better utilize hardware capabilities
>
> Regarding LLM tasks: We are currently assessing UKAN for tasks in protein/molecular science as follow-up work, mainly in the context of diffusion models for proteins and ligands. Our preliminary evaluations suggest UKAN/KAN can match MLP performance in classification tasks, indicating potential for exploring UKAN as a component in larger language models. We conducted fine-tuning tasks on the ESM2 model with the UKAN classifier and results were matching MLP results.
>
> # Weaknesses
>
> We will enhance the paper with:
> 1. Clearer explanations of grid size and parameters
> 2. Detailed parameter definitions in equations
>
> Thank you for helping us improve the clarity and completeness of our paper.

---

> > ### Comment · Reviewer_YkyW · 2024-11-26
> >
> > Thanks for the response. I will keep my score unchanged.

---

### Official Review · Reviewer_Pik8 · 2024-11-01

**Soundness:** 3
**Presentation:** 2
**Contribution:** 2
**Rating:** 5
**Confidence:** 2

**Summary:**

The paper introduces Unbounded Kolmogorov-Arnold Networks (UKANs), a novel approach to function approximation that addresses the limitations of traditional Kolmogorov-Arnold Networks (KANs), specifically the need for bounded grids and inefficiencies in computation and memory usage. UKANs utilize a coefficient generator (CG) model, which dynamically generates B-spline coefficients over an infinite grid, integrating multilayer perceptrons (MLPs) with KANs and leveraging positional encoding for efficient large-scale learning. The authors present a GPU-accelerated implementation that significantly reduces the computational cost of B-spline evaluations. Experimental results across regression, classification, and generative tasks show the effectiveness of UKANs, demonstrating superior computational efficiency and competitive accuracy compared to existing methods. The work advances function approximation techniques, offering a scalable and flexible solution for complex tasks.

**Strengths:**

The authors provide a significant contribution in making function approximation more scalable and efficient. The integration of a coefficient generator (CG) model that dynamically produces B-spline coefficients enables UKANs to handle unbounded domains, a major advancement over existing KAN architectures. The use of GPU acceleration to reduce computational and memory costs is another strong aspect, as it makes UKANs practical for large-scale applications that were previously out of reach for KANs.

**Weaknesses:**

- As shown in https://www.arxiv.org/abs/2407.16674, KAN can be considered a more interpretable model, particularly effective when applied to symbolic formulas. So improving the performance on downstream tasks may not be that important.
- Additionally, as shown in Tables 3 and 4, the performance improvements reported for UKAN compared to KAN are not substantial. These improvements could simply be due to the increased number of parameters or some level of randomness in the training process. The authors should consider evaluating UKAN on a broader range of datasets to strengthen the claims about its effectiveness.
- A few minor suggestions: making Figure 3 smaller and Figure 4 larger would improve readability.
- Also, I noticed a small typo in the caption of Figure 3, where the KAN paper is cited twice, and the use of ‘[32, 32]’ appears unnecessarily. (I’m not sure why the manuscript does not have line numbers.)

**Questions:**

- Have you thought about evaluating UKAN on more diverse datasets to provide a stronger and more convincing comparison, particularly in scenarios where interpretability is crucial?
- What are your thoughts on the potential interpretability trade-offs between KAN and UKAN, given that KANs are known to be more interpretable for symbolic tasks?

---

> ### Author Response · Authors · 2024-11-18
> **Response to Reviewer 2 (Pik8)**
>
> We appreciate the reviewer's insights regarding interpretability and the comparison with symbolic regression tasks.
>
> # Regarding interpretability:
>
> There exists a one-to-one mapping between UKAN and KAN with specific grid bounds. After training, one can evaluate the CG model for all inputs and store the results as KAN parameters. This transformation enables the application of all symbolic regression and pruning techniques used in KAN, preserving interpretability while gaining benefits of UKAN during training.
>
> We plan to extend UKAN/KAN to practical applications in chemistry and biology, including molecule generation and docking. Our experiments demonstrate that combining MLP with UKAN/KAN improves performance in generative tasks.
>
> # Weaknesses
>
> Regarding the arxiv.org/abs/2407.16674 comparison: While this work attempts a fair comparison based on parameter count, we believe a truly fair comparison should consider optimized implementation FLOPs not code-based FLOPs. Publicly available KAN's FLOPs scale with grid size, whereas our warpKAN implementation significantly reduces this overhead. While MLPs benefit from decades of optimization in generalized matrix multiplication, KAN optimization is still in its early stages. Please see the response to the third reviewer for the performance gap between MLP and KAN/UKAN.
>
>
> We have addressed the formatting suggestions:
> - Removed the confusing [32, 32] notation from Figure 3
> - Adjusted Figures 3 and 4 for better readability
>
> ## Thank you again for engaging in our work! If you have any further concerns or questions, we will be happy to address them.

---

### Official Review · Reviewer_fRP4 · 2024-11-03

**Soundness:** 3
**Presentation:** 2
**Contribution:** 2
**Rating:** 6
**Confidence:** 3

**Summary:**

This paper provides a GPU implementation of KAN that utilizes local matrix representations of B-spline functions. In addition, the paper proposes using MLPs to generate B-spline coefficients by embedding the grid-group index and feature index. Experiments are conducted to compare the performance of competing methods in terms of computational efficiency and the accuracy of regression, classification, and generative tasks.

**Strengths:**

The proposed wrapKAN reduces computation time compared to the original torchKAN. The proposed UKAN offers performance that is either better or comparable in regression, classification, and generative tasks.

**Weaknesses:**

The primary contribution of this paper is the introduction of unbounded grid. However, the advantages of the unbounded grid are called into question. Why is grid updating or data normalization in KAN not considered preferable? The experimental results indicate that the improvements of UKAN over KAN are limited.

**Questions:**

If the grids are unbounded, it appears that an infinite number of coefficients would be required for B-spline curves. Is this the case? If not, how does it different from grid updating?

---

> ### Author Response · Authors · 2024-11-18
> **Response**
>
> We thank the reviewer for their thorough feedback and valuable suggestions. Below we address reviewer's concerns:
>
> # Answer for Weaknesses
>
> The grid updating approach in KAN faces several technical challenges:
> 1. It introduces numerical instability during training
> 2. It requires data transfer between GPU and CPU to maintain stability
> 3. These limitations prevent KAN with grid updates from scaling to larger datasets and model sizes
>
> In our case studies, such as MNIST classification, UKAN demonstrates consistent improvements over KAN (96.29% vs 95.35% validation accuracy), verified through independent training runs to ensure statistical significance.
>
> # Answer to Question
>
> Our UKAN approach solves these issues by using a coefficient generator (CG) model that generates coefficients on-the-fly. While theoretically supporting infinite coefficients, UKAN only generates 2K+2 parameters per input, significantly reducing memory overhead compared to storing and updating both grid and coefficient parameters in traditional KAN.
>
> ## Thank you again for reviewing our work! If you have any further concerns or questions, we will be happy to address them.

---

> > ### Comment · Reviewer_fRP4 · 2024-12-01
> >
> > Thank you for the feedback. I believe that effectively determining the grid and improving computational efficiency are two important issues in KAN, and I appreciate that this paper has focused on these aspects. However, the algorithm details are not clearly presented. I would like to increase my score to 6—not higher, as I am not entirely convinced about the ease of implementation of the current Warp UKAN model.

---

### Meta-Review · Area_Chair_1UNL · 2024-12-20

**Metareview:**

This paper extends KANs networks to unbounded grids and proposes a fast(er) library.

In my opinion the weaknesses of this paper outweighs its strengths:
- The performance improvements reported for UKAN compared to KAN do not seem to be significant.
- The presentation can be significantly improved to showcase the contributions:
  - More details on the KAN architecture (instead of referring to another paper) in order to emphasize the differences between KAN and UKAN
  - More details on the B-splines (the constants $k$, $d_{in}$,... are important to appreciate the improvements)
  - More details on how you managed to get a faster implementation than the baselines

**Additional Comments On Reviewer Discussion:**

Reviewer fRP4 provided an initial rating of 5 with a very short review and eventually moved their rating to 6 after the discussion.

I believe Reviewer Pik8 and Reviewer YkyW provided more thorough reviews that I weighted more in my decision.

---

### Decision · Program_Chairs · 2025-01-22

Reject